# Psychosocial Factors Associated with Memory Complaints during the First Wave of the COVID-19 Pandemic: A Multi-Country Survey

**DOI:** 10.3390/brainsci13020249

**Published:** 2023-01-31

**Authors:** Morenike Oluwatoyin Folayan, Roberto Ariel Abeldaño Zuñiga, Jorma I. Virtanen, Oliver C. Ezechi, Nourhan M. Aly, Joanne Lusher, Annie L. Nguyen, Maha El Tantawi

**Affiliations:** 1MEHEWE Study Group, Obafemi Awolowo University, Ile-Ife 220282, Nigeria; 2Department of Child Dental Health, Obafemi Awolowo University, Ile-Ife 220282, Nigeria; 3The Centre for Reproductive and Population Health Studies, Nigerian Institute of Medical Research, Yaba, Lagos 101212, Nigeria; 4Postgraduate Department, University of Sierra Sur, Oaxaca 70805, Mexico; 5Faculty of Medicine, University of Bergen, 5020 Bergen, Norway; 6Department of Pediatric Dentistry and Dental Public Health, Faculty of Dentistry, Alexandria University, Alexandria 21527, Egypt; 7Provost’s Group, Regent’s University, London E14 2BE, UK; 8Department of Family Medicine, Keck School of Medicine, University of Southern California, Los Angeles, CA 91803, USA

**Keywords:** social isolation, emotional distress, SARS-CoV-19, social support, financial support, memory disorders, amnesia, neurodegenerative disorder

## Abstract

This study assessed the associations between psychosocial factors (social isolation, social support, financial support and emotional distress) and memory complaints during the COVID-19 pandemic. This was a secondary analysis of data extracted from the dataset of participants recruited from 151 countries for a COVID-19 related mental health and wellness study between June and December 2020. The dependent variable was memory complaint, measured using the Memory Complaint Questionnaire. The independent variables were perception of social isolation, social support, financial support, emotional distress and history of SARS-CoV-19 infection. Confounding variables were age, sex at birth, level of education, employment status, HIV status and country-income level. Multivariable logistic regression was used to determine the associations between the dependent and independent variables after adjusting for the confounders. Of the 14825 participants whose data was extracted, 2460 (16.6%) had memory complaints. Participants who felt socially isolated (AOR: 1.422; 95% CI: 1.286–1.571), emotionally distressed (AOR: 2.042; 95% CI: 1.850–2.253) and with history of SARS-CoV-19 infection (AOR: 1.369; 95% CI: 1.139–1.646) had significantly higher odds of memory complaints. Participants who perceived they had social and financial support had significantly lower odds of memory complaints (AOR: 0.655; 95% CI: 0.571–0.751). Future management of pandemics like the COVID-19 should promote access to social and financial support and reduce the risk of social isolation and emotional distress.

## 1. Introduction

Cognitive problems appear to be a consistent feature of COVID-19, with up to a fifth of patients with COVID-19 having memory complaints [1]. Cognitive problems are associated with mild, moderate and severe forms of COVID-19, with prevalence increasing with the severity of COVID-19 [2] where the infection affects the nervous system [3]. Patients with COVID-19 seem to lose grey matter volume, thereby increasing the risk of impaired processing of information in the brain, and the impact on the management of emotions, memories and movements [4,5,6]. There is also the possibility of trans-synaptic viral spread to cortical regions, including the hippocampus, following invasion of the peripheral olfactory neurons with a negative impact on the spatial and episodic memory [7,8,9,10,11]. 

There may also be other factors associated with cognitive problems during the COVID-19 infection. For instance, COVID-19 causes critical illnesses that lead to delirium and acute respiratory distress syndrome, both of which are associated with cognitive impairment [12]. Also, the hypoxia and elevated glucocorticoid concentrations associated with COVID-19 infection make the brain, especially the hippocampus, more vulnerable to damage that affects memory [13,14]. 

A number of sociodemographic factors are associated with a higher risk of cognitive problems due to COVID-19. These include lower education and social support [15,16,17]. Low education could lead to limited memory capacity, while people with high education levels could maintain or increase their cognitive functional development through frequent stimulation of the brain during daily life [18,19]. Unemployment also limits access to employment-induced cognitive health benefits [20]. Also, low social support reduces mental stimulation and decreases cerebral neuronal growth, thereby expediting cognitive decline [21]. In addition, low social support reduces active participation in community activities and increases the risk of cognitive problems [22,23]. Furthermore, emotion has been inextricably linked to cognitive processes [24]. For example, depression increases the risk of cognitive impairment [25], as do other forms of emotional stress [26]. Cognitive decline is also associated with HIV infection as the disease progresses [27].

Although several studies have reported multiple symptoms that persist after recovery from COVID-19 infection, there are few studies reporting the cognitive impairment associated with the disease. One of the earliest assessments is memory complaints. Memory complaints are early symptom neurodegenerative disorders, occurring on the pathway to cognitive decline and dementia [28,29,30]. This study attempts to address this gap in knowledge by investigating factors that might be associated with memory complaints during the COVID-19 pandemic. Specifically, the present study assessed the associations between psychosocial factors (social isolation, social support, financial support and emotional distress) and memory complaints during the COVID-19 pandemic. It was hypothesised that social isolation and emotional distress would be directly associated with memory complaints, while access to social and financial support would be inversely associated with memory complaints.

## 2. Materials and Methods

Ethical approval for the study was obtained from the Human Research Ethics Committee at the Institute of Public Health of the Obafemi Awolowo University Ile-Ife, Nigeria (HREC No: IPHOAU/12/1557), Brazil (CAAE N° 38423820.2.0000.0010), India (D-1791-uz and D-1790-uz), Saudi Arabia (CODJU-2006F) and United Kingdom (13283/10570) for the conducting of the primary study. Participants checked a box to indicate consent before participating in the online survey.

### 2.1. Sample Size

This primary study recruited 21,106 participants from 152 countries between July and December 2020 through an online survey. Participation was open to anyone 18 years and above if they could access the survey over the internet using an electronic device and if they could understand the languages of the survey (English, French, Spanish, Arabic and Portuguese). There were no exclusion criteria. The sample size was considered adequate as it was set at 35 valid respondents from each of the 193 member States of the United Nations. The sample size was increased by 10% because of the risk of missing responses in the absence of guidance, support and motivation for survey response when collecting data online [31]. Online data collection was carried out in view of the restrictions during the first wave of the COVID-19 pandemic when these data were collected.

From this study, the extracted data of 14,825 participants (70.2% of the dataset of the primary study) were considered adequate for statistical modelling since a minimum of 10 participants with complete responses per dependent variable existed. This enabled the performance of regression analyses with a minimum probability level (*p*-value) of 0.05 [32]. 

### 2.2. Recruitment Procedure

Details of the study, including the recruitment process, have been previously published [33,34,35,36]. Non-probability sampling was employed with recruitment driven by the 45 members of the MEHEWE Study group (www.mehewe.org (accessed on 24 December 2022)). The survey link was shared with contacts around the world using social media platforms (Facebook, Twitter, and Instagram), network email lists and WhatsApp groups. Details concerning the conducting of the survey and the data collection tools are published elsewhere [35].

### 2.3. Data Collection Tools

In brief, the data collection tool was validated using both quantitative and qualitative assessments [35]. The instrument was first developed in English and translated into French, Spanish, Arabic and Portuguese. The translations were back-translated to English to ensure that they retained their meaning. The overall content validation index for the questionnaire was 0.83. The dimensionality and reliability of the tool was also assessed. The details on the validation of the data collection tool had also been published elsewhere [35].

Data were collected anonymously. The privacy of participants and the confidentiality of the information provided was protected by decoupling the IP addresses from the questionnaire at the end of the online survey. The questionnaire also did not install any tracker cookies on the devices of the respondents. Data were collected using SurveyMonkey^®^ which provides a secured, SSL encrypted connection link. Data in transit (while responding online) were encrypted using secure TLS cryptographic protocols. The collection tool was certified in compliance with the EU-U.S. Privacy Shield Framework and Swiss-U.S. Privacy Shield. 

### 2.4. Dependent Variable

Data were collected on memory complaints using the Memory Complaint Questionnaire [37] that had been validated for use as a self-reported memory questionnaire. The tool consists of six questions on memory functioning in daily circumstances. Participants were asked to compare and evaluate their current performance to that before the COVID-19 pandemic. The total score ranges from 7 to 35, with higher values indicating subjective memory loss. Scores higher than or equal to 25 are indicative of memory impairment. Participants were grouped into those without significant memory complaints (Memory Complaint Questionnaire scores of <25) and those with significant memory complaints (Memory Complaint Questionnaire scores of ≥25) [37]. The content validity index (CVI) for the section of the questionnaire was 0.90, the ICC was 0.71 and the Cronbach alpha score was 0.94 [35].

### 2.5. Independent Variables

*Social isolation, social support and financial support*: Participants were asked to identify how socially isolated they felt compared to before the COVID-19 pandemic. Response options were the same, less socially isolated, more socially isolated. The social isolation variable was dichotomised into same/less socially isolated versus more socially isolated. Also, participants were asked about their perceived access to social and financial support during the COVID-19 pandemic. Response options were ‘yes’ or ‘no’. These questions were adopted from the Coronavirus Health Impact Survey (CRISIS) Adult Self-Report Baseline questionnaire [38]. The CVI for this section of the questionnaire was 0.90, the ICC was 0.89 and the Cronbach alpha score was 0.93 [35].

*Emotional distress*: Participants were asked if they had experienced any form of emotional distress (frustration or boredom, anxiety, depression, loneliness, anger and grief/ feeling of loss) during the pandemic by checking a box against the emotions experienced. Respondents who did not check a response were categorised as not having emotional distress during the pandemic. The CVI for this section of the questionnaire was 0.90 [35].

*SARS-CoV-19 infection*: Participants were asked to identify if they had had a SARS-CoV-19 infection by ticking a checkbox. A tick of the checkbox was an indication of having a history of SARS-CoV-19 infection (yes). All those who did not tick the box were categorised as not having had SARS-CoV-19 infection at the time of the survey (no).

### 2.6. Confounders

*Sociodemographic variables*: Data were extracted about age at last birthday; sex at birth (male, female and others dichotomised into male and non-male), level of education (no formal education, primary, secondary and college/university), and employment status (retiree, student, employed and unemployed).

*HIV status*: Participants identified their HIV status by checking off a list of 27 medical ailments. A tick on the checkbox for HIV was an indication that the individual was living with HIV. The list of medical ailments was adopted from Marg et al. [39]. The CVI for the section of the questionnaire that contained details on the HIV status during the pandemic was 0.71 [35].

*Country income level*: Information about the country income level was obtained from publicly available data of the World Bank Data Bank [40]. Countries were classified into low-income countries (LIC) with a gross national income (GNI) per capita ≤ 1035 USD in 2019, lower-middle-income countries (LMIC) with GNI between 1036 and 4045 USD, upper-middle-income countries (UMIC) with GNI between 4046 and 12,535 USD and high-income countries (HIC) with GNI ≥ 12,536 USD.

### 2.7. Data Analysis

Raw data were downloaded, cleaned, and imported to SPSS version 23.0 (IBM Corp., Armonk, NY, USA) for analyses. A description of the variables was conducted. Bivariate analysis included comparing participants with and without memory complaints regarding the confounders and independent variables using chi squared test (and t test for age), followed by an estimation of effect size using Phi squared (and r squared for age). Squared values of 0.01–0.09 indicate small effect sizes, 0.10–0.24 indicate medium effect sizes, 0.25 to 0.49 indicate large effect sizes and greater than 0.49 indicate very large effect sizes [41,42]. Also, a multivariable regression analysis was used to determine the associations between the dependent and independent variables after adjusting for the confounders. Adjusted odds ratios (AOR) and 95% confidence intervals (CI) were calculated. Statistical significance was set at 5%.

## 3. Results

Table 1 shows that the 14,729 participants had ages ranging from 18–99 years and a mean (standard deviation) age of 35.3 (12.8) years. There were 9222 (62.6%) females, 11568 (78.0%) with college/university level of education, 8625 (58.2%) employed and 7845 (52.9%) living in high-income countries at the time of collecting the data. 

Of the 14,729 participants, 2446 (16.6%) reported memory complaints. People who reported memory complaints were 1747 (19.6%) of 8934 participants who reported social isolation, 2093 (15.6%) of 13,381 who perceived they had social and financial support, 1680 (21.7%) of 7747 participants who felt emotionally distressed, and 160 (22.5%) of the 736 participants who had a history of SARS-CoV-19 infection.

Participants who reported social isolation (AOR: 1.376; 95% CI: 1.243–1.522; *p* < 0.001), emotional distress (AOR: 2.071; 95% CI: 1.875–2.288; *p* < 0.001) and who had a history of SARS-CoV-19 infection (AOR: 1.394; 95% CI: 1.154–1.681; *p* = 0.001) had significantly higher odds of memory complaints than participants who did not feel socially isolated, who did not feel emotionally distressed and who did not have SARS-CoV-19 infection, respectively. By contrast, participants who perceived they had social and financial support had significantly lower odds of memory complaints than participants who did not have social and financial support (AOR: 0.655; 95% CI: 0.571–0.751; *p* < 0.001). 

Table 1 also shows that the Phi^2^ coefficients indicating effect size for social isolation, social and financial support, emotional distress and SAR-COV-19 infection were minuscule. (Phi^2^ = 0.01, 0.01, 0.02 and 0.001).

## 4. Discussion

In support of the hypotheses, the results of this current study suggest that people who felt socially isolated, emotionally distressed and who had a history of SARS-CoV-19 infection during the COVID-19 pandemic were more likely to experience memory complaints than those who did not experience any of these. Also, access to social and financial support during the pandemic reduced the risk of memory complaints.

One strength of this study is that it provides further evidence of a relationship between psychosocial factors and mental health [43]. It also contributes to the evolving evidence that memory complaints are associated with the COVID-19 pandemic. Moreover, this study generates new evidence from a large global sample. Nevertheless, findings do need to be considered in the light of the cross-sectional nature of the design, and respect that direct cause-inferential deductions cannot be inferred from these results. The study participants were also recruited online inadvertently making those without smartphones and internet access ineligible for participation. In addition, the survey was conducted in only a few languages, thereby excluding those who do not understand the languages the survey was conducted in. These factors thereby limit the generalisability of the findings to some extent. In addition, confounding variables such as comorbidities and types of medications that can affect memory were not adjusted for. We were unable to conduct these adjustments because the relevant information was not available in the dataset. Furthermore, the measure for emotion stress was carried out using a respondent-rated single-item question. Single-item measures of emotional stress such as depression have, however, been found to be highly specific and appropriate for ruling out cases. Single-item questions, however, have low sensitivity with implications for the underestimation of the cases of emotional stress in this study cohort [44]. We also acknowledge that culture may affect the way information is filtered into the memory and can affect memory specificity and memory resolution for previously-encoded items [45]. Cultural differences were not adjusted for as a confounder in this study. Despite these limitations, the study does generate new and useful information that serves as leading to more interesting hypotheses that could go on to further inform program planning in the management of future pandemics. 

Psychosocial factors are considered important in the aetiology of mental health problems. Mental health problems such as memory impairment may result from the interplay of several variables, including environmental stressors, personal and environmental resources and the individual’s appraisal and coping responses to specific stressful events [46]. Several models have described how mental health outcomes may be affected by psychosocial factors including emotional, behavioural and physiological stressors contributing to strain and poor health [47]. The present study findings suggest that psychosocial factors may affect mental health through pathways that promote cognitive degeneration. Psychosocial factors may cause cognitive degeneration through multiple neurobiological mechanisms such as those associated with cerebral infarction, and neurodegenerative pathologies. Other pathways may be independent of the traditional pathological pathways [48].

We observed that social isolation is associated with memory complaints during the COVID-19 pandemic. Prior studies had identified that social isolation has a detrimental effect on the memory because of a lack of social stimulation on the brain, resulting from low levels of social contacts [49,50]. This leads to lower cognitive reserve, poorer resilience of the brain and cognitive impairment [51,52,53,54]. Social isolation may also induce emotional distress [55]. The COVID-19 pandemic contributed significantly to social isolation [56], emotional distress [57] and high risk of poor access to social and financial support, especially for those who had COVID-19 [58]. Prior studies raised concerns about the negative impact of lockdown during the COVID-19 pandemic on feelings of isolation leading to emotional distress, health problems and early mortality [59]. Others have argued against isolation on ethical grounds [60]. We provide suggestive evidence here that the lockdown and accompanying feelings of social isolation and emotional distress may actually contribute to memory complaints. The study, however, did not preclude the possibility that memory decline may have led to social isolation as this is another plausible interpretation of the direction of events [61]. Further studies are necessary to elucidate the direction of effect between social isolation and emotional distress, over time, though this is likely to be somewhat multidirectional, with no clear pathway. The study results, however, indicate that the effect size of social isolation and emotional distress on memory complaints is minuscule, indicating an extremely weak psychological effect.

As the study findings indicate, access to social and financial support during the COVID-19 pandemic appeared to be associated with lower odds of memory complaints. It is known that social support improves mood and reduces the risk of cognitive impairment [62]. Social support has also been shown to provide a buffer against functional decline for people with depression [63]. The study results, however, indicate that the effect size is minuscule, indicating an extremely weak psychological effect. The pathway is, however, poorly studied and further studies are needed to better understand the neurobiological mechanisms that underpin this phenomenon.

Ongoing studies on the association between SARS-CoV-19 infection and memory complaints have indicated that there are neurobiological mechanisms linking the two phenomena [7,8,9,10,11,12,13,14]. Our study findings extend and reinforce prior evidence of an association between SARS-CoV-19 infection and memory complaints, although the effect size is minuscule. The evidence so far suggests that mental health support in the form of cognitive health care should be instituted for all those with SARS-CoV-19 infection. Cognitive health care promotes the ability to clearly think, learn and remember, which supports the effective performance of everyday activities [64]. 

## 5. Conclusions

In conclusion, social isolation, emotional distress and a history of SARS-CoV-19 infection were likely possible risk factors for memory complaints during the COVID-19 pandemic, though their effects on memory complaints were very small. Access to social and financial support seems to be associated with a lower risk of memory complaints. The future management of pandemics like COVID-19 should promote access to social and financial support and reduce the risk of social isolation and emotional distress.

## Figures and Tables

**Table 1 brainsci-13-00249-t001:** Multivariable logistic regression analysis to determine factors associated with memory complaints among adults who participated in the global survey (N = 14,729).

Variables	Total N = 14,729 n (%)	Memory Compliant		X^2^	Phi^2^	AOR; 95% CI; *p* Value
		Yes N = 2446 (16.6) n (%)	No N = 12,283 (83.4) n (%)			
**Economic region**				86.88	0.01	
LIC	348 (2.4)	84 (24.1)	264 (75.9)			1.665; 1.273–2.177; *p* < 0.001
LMIC	7795 (52.9)	1096 (14.1)	6699 (85.9)			0.912; 0.813–1.023; *p* = 0.116
UMIC	2977 (20.2)	588 (19.8)	2389 (80.2)			1.048; 0.922–1.191; *p* = 0.470
HIC	3609 (24.5)	678 (18.8)	2931 (81.2)			1.000
**Level of education**				38.55	0.003	
None	294 (2.0)	79 (26.9)	215 (73.1)			3.114; 2.336–2.741; *p* < 0.001
Primary	361 (2.5)	85 (23.5)	276 (76.5)			2.102; 1.612–2.741; *p* < 0.001
Secondary	2568 (17.5)	400 (15.6)	2168 (84.4)			1.076; 0.947–1.223; *p* = 0.262
College/university	11,506 (78.0)	1882 (16.4)	9624 (83.6)			1.000
**Employment status**				137.15	0.01	
Retired	569 (3.9)	168 (29.5)	401 (70.5)			1.412; 1.092–1.826; *p* = 0.008
Student	3270 (22.2)	395 (12.1)	2875 (87.9)			1.071; 0.901–1.273; *p* = 0.438
Employed	8578 (58.2)	1552 (18.1)	7026 (81.9)			1.431; 1.244–1.646; *p* < 0.001
Unemployed	2312 (15.7)	331 (14.3)	1981 (85.7)			1.000
**Age**	35.3 (12.8)	38.8 (13.2)	34.6 (12.6)	-	0.027 ^¶^	1.021; 1.017–1.026; *p* < 0.001
**Sex at birth**				44.97	0.003	
Male	5507 (37.4)	768 (13.9)	4739 (86.1)			1.000
Female	9222 (62.6)	1678 (18.2)	7544 (81.8)			1.395; 1.266–1.358; *p* < 0.001
**HIV positive**				0.22	<0.0001	
Yes	905 (6.1)	155 (17.1)	750 (82.9)			0.915; 0.751–1.114; *p* = 0.377
No	13,824 (93.9)	2291 (16.6)	11,533 (83.4)			1.000
**Socially isolated**				142.12	0.01	
Yes	8934 (60.7)	1747 (19.6)	7187 (80.4)			1.376; 1.243–1.522; *p* < 0.001
No	5795 (39.3)	699 (12.1)	5096 (87.9)			1.000
**Social and financial support**				99.35	0.01	
Yes	13,381 (90.8)	2093 (15.6)	11,288 (84.4)			0.630; 0.548–0.723; *p* < 0.001
No	1348 (9.2)	353 (26.2)	995 (73.8)			1.000
**Emotional distress**				306.02	0.02	
Yes	7747 (52.6)	1680 (21.7)	6067 (78.3)			2.071; 1.875–2.288; *p* < 0.001
No	6982 (47.4)	766 (11.0)	6216 (89.0)			1.000
**SARS-CoV-19 infection**				19.59	0.001	
Yes	736 (5.0)	160 (22.5)	576 (78.3)			1.394; 1.154–1.681; *p* = 0.001
No	13,993 (95.0)	2286 (16.3)	11,707 (83.7)			1.000

AOR: adjusted odds ratio, CI: confidence interval; ^¶:^ r2 used for effect size instead of Phi^2^.

## Data Availability

Data are available upon request.

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
