# Peer review of "Psychosocial Factors Associated with Memory Complaints during the First Wave of the COVID-19 Pandemic: A Multi-Country Survey"

_brainsci, 2023, doi:10.3390/brainsci13020249_

Round 1
Reviewer 1 Report
The paper adds to the base of knowledge. however, I think you need to add the correlation between memory complains and other socio-demographics factors since its already mentioned in the introduction but nothing was related in the results.
The paper presentation is good, however, the conjunction words is to connect between sentences in the same paragraph. so we do not use it at the beginning of new paragraph. such as furthermore, however, ....etc.
- You did not mentioned the characteristics of the sample, I know its already mentioned in previous paper, however you need to give brief summary here about it.
- You did not do correlation between your independent variable and the socio-demographics of participants.
- since there were no exclusion criteria there were many con-founding factors affect your findings such as comorbidity and type of disease, type of medication.
- emotional stress measuring scale did not reflect the actual emotional problems since most of people could not differentiate between stress anxiety, depression....etc, additionally, the culture play a role in understanding and personal perception of these emotional problems.
It was preferable to use a valid scale, e.g DASS to measure these variables. therefore, you need to mention all of the above as limitations of your study, transcultural variations was the main limitation of your study. based on these limitation you have to recommend future studies.
- The limitations of study usually mention at the end of the discussion not at the beginning.
- The implications of the study need to be highlighted
Author Response
Thanks for the feedback on the manuscript. We found the comments extremely useful. We feel the quality of the manuscript has improved significantly with the comments received. In view of one of the comments received, we re-ran the analysis to include age and sex a birth. The values of the AOR and CI changed but the level of significance remained similar for all the variables.
The paper adds to the base of knowledge. however, I think you need to add the correlation between memory complains and other socio-demographics factors since its already mentioned in the introduction but nothing was related in the results.
Response: Thanks a million for this suggestion. My assumption is that the reviewer will like us to include a tabulation of the tests of association using bivariate analysis (t-tests and chi-square). This analysis is not of value for the objective of the study as we did not do a selection of variables for the development of the logistic regression analysis. We have therefore decided to leave out building a table that shows the correlation analysis
The paper presentation is good, however, the conjunction words is to connect between sentences in the same paragraph. so we do not use it at the beginning of new paragraph. such as furthermore, however, ....etc.
Response: Thanks for raising this. We have read through the manuscript and addressed this suggested edit. We made edits to three paragraphs with this gap noted.
- You did not mentioned the characteristics of the sample, I know its already mentioned in previous paper, however you need to give brief summary here about it.
Response: Thanks for identifying this need. We wrote: Table 1 shows 14729 participants with age ranging from 18- 99 years and a mean (standard deviation) age of 35.3 (12.8) years. There were 9222 (62.6%) females, 11568 (78.0%) with college/university level of education, 8625 (58.2%) employed and 7845 (52.9%) living in high income countries at the time of collecting the data.
- You did not do correlation between your independent variable and the socio-demographics of participants.
Response: Thanks for raising this. We left out the correlation analysis as this was not a study objective. It also was not of value to help us build the regression model.
- since there were no exclusion criteria there were many con-founding factors affect your findings such as comorbidity and type of disease, type of medication.
Response: We feel this is very important. We have included this as a study limitation. We could not control for this as this was a secondary data analysis. We wrote: In addition, confounding variables like comorbidities and types of medications can affect memory were not adjusted for. We were unable to conduct these adjustments because the relevant information were not available in the dataset
- emotional stress measuring scale did not reflect the actual emotional problems since most of people could not differentiate between stress anxiety, depression....etc, additionally, the culture play a role in understanding and personal perception of these emotional problems.
Response: Thanks for raising this limitation. We agree with the observation and have included these as limitations for the study. We note: In addition, confounding variables like comorbidities and types of medications can affect memory were not adjusted for. We were unable to conduct these adjustments because the relevant information were not available in the dataset. Furthermore, the measure for emotion stress was done using a respondent-rated single-item question. Single item measures of emotional stress like depression have been found to be highly specific though and appropriate for ruling out cases. Single-item questions however, have low sensitivity with implications for the underestimation of the cases of emotional stress in this study cohort [42]. We also acknowledge that culture may affect the way information is filtered into the memory and can affect memory specificity and memory resolution for previously-encoded items [43]. Cultural differences were not adjusted for as a confounder in this study.
It was preferable to use a valid scale, e.g DASS to measure these variables. therefore, you need to mention all of the above as limitations of your study, transcultural variations was the main limitation of your study. based on these limitation you have to recommend future studies.
Response: Thanks for raising this. We have acknowledged the limitation with the use of a single-question measure of emotional stress in the study limitations.
- The limitations of study usually mention at the end of the discussion not at the beginning.
Response: We have adopted the style suggested by STROBE. Limitations come early so that readers are able to have information on the biases associated with the study design and then, engage with the discussion through the lens of this biases.
- The implications of the study need to be highlighted
Response: We had an implication for the study finding that said: The evidence so far suggests that mental health support in the form of cognitive health care should be instituted for all those with SARS-CoV-19 infection. Cognitive health care promotes the ability to clearly think, learn, and remember which supports the effective performance of everyday activities [62].

Reviewer 2 Report
The aim of the present study assessed the association between psychosocial factors and memory complaints during COVID-19 pandemic. The authors bring up an interesting topic and timely study. The manuscript is well written in an engaging and lively style.
Point 1: Is there any explanation as to why 152 countries were selected? Is it a case where there are more than 35 respondents out of 193 UN countries? If so, wouldn't the response rate be low if these languages (English, French, Spanish, Arabic and Portuguese) are not native languages? For example, East Asian countries such as Japan and Korea are likely to have very low response rates.
Point 2: Generally, the strengths and limitations of the study appear at the end of the discussion. Is there a reason why you wrote it at the beginning of the discussion? Also, a more detailed description and numbering of the strengths and limitations of the study will make it easier for readers to read.
Author Response
Thanks for the feedback on the manuscript. We found the comments extremely useful. We feel the quality of the manuscript has improved significantly with the comments received. In view of one of the comments received, we re-ran the analysis to include age and sex a birth. The values of the AOR and CI changed but the level of significance remained similar for all the variables.
REVIEWR 2
The aim of the present study assessed the association between psychosocial factors and memory complaints during COVID-19 pandemic. The authors bring up an interesting topic and timely study. The manuscript is well written in an engaging and lively style.
Response: Thanks for the positive feedback
Point 1: Is there any explanation as to why 152 countries were selected? Is it a case where there are more than 35 respondents out of 193 UN countries? If so, wouldn't the response rate be low if these languages (English, French, Spanish, Arabic and Portuguese) are not native languages? For example, East Asian countries such as Japan and Korea are likely to have very low response rates.
Response: The aim was to collect data globally. We feel that the limited number of languages for the data collection tool may have limited our ability to cover the 193 UN countries. We acknowledged this in the primary study.
Point 2: Generally, the strengths and limitations of the study appear at the end of the discussion. Is there a reason why you wrote it at the beginning of the discussion? Also, a more detailed description and numbering of the strengths and limitations of the study will make it easier for readers to read.
Response: Thanks for the question. We have adopted the style suggested by STROBE. Limitations come early so that readers are able to have information on the biases associated with the study design and then, engage with the discussion through the lens of this biases. We have also raised a number of new limitations based on the first reviewer’s comments.
